# The Effects of Filter Ventilation and Expanded Tobacco on the Tar, Nicotine and Carbon Monoxide Yields from Cigarettes Sold in Australia

**DOI:** 10.3390/ijerph22010050

**Published:** 2024-12-31

**Authors:** Wendy R. Winnall, Ashleigh Haynes, Walther Klerx, Ingrid M. E. Bakker-‘t Hart, Caroline Versluis, Niels M. Leijten, Emily Brennan, Reinskje Talhout, Melanie A. Wakefield

**Affiliations:** 1Centre for Behavioural Research in Cancer, Cancer Council Victoria, Melbourne, VIC 3002, Australia; ashleigh.haynes@cancervic.org.au (A.H.); emily.brennan@cancervic.org.au (E.B.); melanie.wakefield@cancervic.org.au (M.A.W.); 2Melbourne School of Psychological Sciences, The University of Melbourne, Melbourne, VIC 3002, Australia; 3Center for Health Protection, National Institute for Public Health and the Environment, 3720 BA Bilthoven, The Netherlands; walther.klerx@rivm.nl (W.K.); ingrid.t.hart@rivm.nl (I.M.E.B.-‘t.H.); carolina.versluis@rivm.nl (C.V.); niels.leijten@rivm.nl (N.M.L.); reinskje.talhout@rivm.nl (R.T.)

**Keywords:** tobacco, nicotine, cigarette, filter ventilation, expanded tobacco, tar, carbon monoxide, misperceptions, brand variant, Health Canada Intensive

## Abstract

Cigarette brand variant names and characteristics such as the taste and feel of the smoke can mislead consumers into believing some products are less harmful. We assessed the characteristics of three common cigarette variants sold in Australia, “gold”, “blue” and “red”, to determine which characteristics differed by color, and which affected tar, nicotine and carbon monoxide (TNCO) yields. TNCO yields, physical parameters, expanded tobacco and filter ventilation were measured in cigarette color variants from eight brands. Filter ventilation and expanded tobacco were common across brands and variants. Compared to blue and red variants, gold variants had slightly shorter tobacco rods and greater filter ventilation. Gold variants had lower TNCO when measured using the industry-favored International Organization for Standardization (ISO) protocol. ISO-measured TNCO yields were associated with filter ventilation and tobacco rod length, but not use of expanded tobacco. When measured using the Health Canada Intensive (HCI) protocol, which better emulates human smoking behavior, TNCO emissions were markedly higher, and the emission differences by extent of filter ventilation were minimized, indicating that ISO measurements are misleading. These findings confirm that cigarette color names, and the filter ventilation levels they signify, remain misleading more than a decade after plain packaging eliminated pack colors in Australia, as higher levels of filter ventilation are not associated with reduced TNCO emissions measured using the HCI protocol. Consumer education and communication campaigns could amplify the impacts of Australia’s newly passed tobacco legislation banning color and other variant names that imply reduced harm.

## 1. Introduction

The development of combustible cigarettes with reduced risks to human health has long been promoted by the tobacco industry but never actually achieved [1]. Misperceptions persist among people who smoke, such as that their preferred brands are less harmful [2,3,4]. Changes to various cigarette characteristics, and the promotion by the tobacco industry of these characteristics as reducing risk have contributed to these misperceptions for decades [1,5,6].

Tar levels in the smoke of cigarettes sold in Australia have been measured by smoking machines and published by public health authorities from 1967, as well as printed on cigarette packets until 2006 [7,8]. These measurements used the industry-favored International Organization for Standardization (ISO) protocol that is now known to poorly imitate human smoking behavior [9]. The use of filter ventilation in cigarettes has ensured that so-called low tar cigarettes are palatable and popular with consumers [6].

Filter ventilation refers to tiny holes in the tipping paper that wraps around the filter. When smoking a cigarette with a ventilated filter, additional air is inhaled that dilutes the smoke drawn through the filter. But people who smoke filter ventilated cigarettes generally do not inhale less smoke [10]. They use compensatory smoking (either consciously or subconsciously) in order to achieve their desired nicotine levels from each cigarette. People regulate their own nicotine intake by increasing their smoke intake per cigarette, smoking more cigarettes, or blocking the vents with their fingers or lips, thereby compensating for the dilution of the smoke [11]. Yields of tar, nicotine and carbon monoxide (TNCO) reported by tobacco companies are assessed by smoking machines under standard conditions, without factoring in compensatory smoking.

The adoption of “low tar” filter-ventilated cigarettes by the market has not led to better health outcomes for people who smoke [1,12]. No relationship has been found between the machine-measured tar and nicotine levels and the risks for most categories of tobacco-related diseases [1]. The likely reason for this is compensatory smoking; people who smoke maintain their dose of tar and toxicants when they maintain their dose of nicotine [12].

There is a wide discrepancy between the way cigarettes are smoked by people compared to smoking machines. The protocol used by the tobacco industry to measure TNCO was designed to measure the amount of these components when ‘smoked’ by a machine rather than under realistic conditions [7,9]. The ISO protocol has smoking machines take a 35 mL puff volume of 2 s duration every 60 s [13]. A more intensive smoking protocol (Health Canada Intensive, HCI) was developed in an effort to better imitate human smoking behavior [14]. This HCI protocol has the smoking machine take a 55 mL puff volume of 2 s duration every 30 s [15]. The HCI protocol also occludes all filter ventilation holes, similar to the effect of a person covering the holes with their fingers or lips during smoking. TNCO levels measured by the HCI protocol are consistently estimated to be much higher than when using the ISO protocol for the same cigarettes [14,16].

“Mild”, “light”, “ultra-light” and similar terms were used in variant names for cigarettes estimated by ISO smoking machines to have low TNCO levels [17]. These variant names implied that so-called low tar products would have a low risk of causing disease [5,18,19] and of addiction [2]. Despite some public awareness messaging that filter-ventilated “mild” or “light” cigarettes are no less harmful than full-strength cigarettes, people who used these cigarettes, or who believed their cigarettes had filter ventilation, were more likely to believe they are less harmful [4,5,18,19] and less addictive [2]. Such misperceptions could mean that lighter tasting cigarettes are considered by some an intermediate step or alternative to cessation, leading to deference or avoidance of smoking cessation [20].

The World Health Organization’s Framework Convention on Tobacco Control (FCTC) recommends banning the use of variant names such as “light”, “mild” and “low tar” [21]. The use of these terms was ended in Australia in 2006 under the terms of a litigation settlement between tobacco companies and Australia’s consumer regulatory agency, now known as the Australian Competition and Consumer Commission [8]. Unfortunately, the removal of these variant names did not substantially reduce the misperception that less harsh tasting smoke was less harmful [17,22,23]. This is most likely because in Australia and elsewhere, the tobacco industry responded to these bans by converting to variant names such as “smooth” and “fine” [24], and by maintaining the packet colors commonly associated with different levels of filter ventilation [25]. Examples are red for low ventilation, increasing through blue, gold and white, with high ventilation. Even after the introduction of plain packaging in Australia in 2012, these color variant names remained in contemporary brand variant names (Appendix A) [26]. However, it was not known whether the relationship between color and filter ventilation was still maintained in contemporary cigarettes sold in Australia, nor if filter ventilation remains common in these cigarettes.

The use of expanded tobacco is another characteristic that the tobacco industry claims to make cigarettes less harmful by reducing levels of nicotine and total particulate matter in smoke [27]. Tobacco is expanded using high pressure liquid carbon dioxide or organic solvents such as propane, which increase the size of the tobacco matter without adding much weight [28]. It is likely that the use of expanded tobacco to make lower weight cigarettes became popular in Australia prior to 1999, when tobacco was taxed by weight and tobacco companies focused on producing lower weight products [29]. The extent of use of expanded tobacco in contemporary cigarettes sold in Australia is unknown.

Since the removal of explicit TNCO-related labelling from packs in 2006 and the introduction of plain packaging in 2012, there has been minimal published research reporting on TNCO levels or other characteristics of tobacco products sold in Australia. The broad objective of this study was to understand the characteristics of cigarettes that may contribute to misperceptions of lower risk to inform public messaging and policy development to counteract these misperceptions. The first specific aim of this study was to investigate the characteristics of cigarettes that may be associated with TNCO levels, such as filter ventilation, expanded tobacco and other physical characteristics of the cigarettes. The second aim was to examine whether there remains an association between color variant names, the presence and amount of filter ventilation, and TNCO levels, and whether these relationships differ when ISO versus HCI protocols are used for smoking machine tests. Gold, blue and red color variants were chosen due to being the most common choices for Australian smokers (unpublished data from Cancer Council Victoria).

## 2. Materials and Methods

### 2.1. Samples

Cigarettes were sourced from eight popular brands, each with three brand variants: gold (or orange), blue and red, making a total sample of 24 different brand variants, listed in Appendix A. Cigarettes were purchased in February–April 2023 from major supermarkets and tobacconists (the store types where most tobacco products in Australia are purchased [30]). To minimize differences according to batches or store conditions, multiple packs for each brand variant were sourced from three broad geographic regions (North, West and South–East) located within ~20 km of Melbourne city center, Australia. Two reference cigarettes CM9 [31] and 1R5F [32] were purchased from Körber Technologies GmbH (Hamburg, Germany) and the Center for Tobacco Reference Products, University of Kentucky (Lexington, KY, USA).

### 2.2. Physical Characteristics

A caliper was used to measure the cigarette lengths and cigarette diameters (15 mm from the mouth end of the cigarette filter). Cigarette filters were then removed and separately measured, along with tipping paper lengths. The lengths of the tobacco rods were determined by subtracting the filter lengths from the total lengths of the cigarettes.

### 2.3. Expanded Tobacco Measurement

Percent displacement was used to determine the relative volume increase due to the possible presence of expanded tobacco. A total of 10 g of tobacco was weighed from each brand and each reference cigarette: CM9 (which has no expanded tobacco) and 1R6F (which contains expanded tobacco). If 10 g of tobacco could not be obtained from the cigarettes, the amount of expanded tobacco was calculated with the amount of tobacco available. The tobacco was transferred into a 250 mL measuring cylinder and loaded with a rod weighing 1875 ± 5 g with a diameter of 37.3 ± 0.5 mm. After 5 ± 0.2 min, the volume was read without removing the rod. The difference in volume relative to the average volume of CM9 was determined and expressed as the relative volume increase.

### 2.4. Filter Ventilation

Filter pressure drop (open and closed) and filter ventilation were measured in the 24 cigarette brand variants and reference cigarette 1R6F. Filter pressure drop and filter ventilation were measured according to ISO 6565:2015 [33] and ISO 9512:2019 [34], respectively. Pressure drop was measured by a Quantum Neo Solo V instrument (Cerulean, Milton Keynes, UK) and used for calculation of filter ventilation.
$$Degree\;of\;filter\;ventilation\;\left(V_{F}\right)=\frac{Filter\;ventilation\;\left({Q^{\prime }}_{F}\right)}{Total\;air\;flow\;\left(Q\right)}\times 100\%\;$$

Filter ventilation is expressed as the percent of total air flow rate that has entered through the tipping paper when ventilation holes are not occluded. As a quality control sample, 1R6F reference cigarettes were measured and the results were within certified values.

### 2.5. Tar, Nicotine and Carbon Monoxide (TNCO)

Sample cigarettes and control CM9 [31] reference cigarettes were conditioned as described in ISO 3402:2023 [35]. Cigarettes were smoked in a 20-port rotary smoking machine (RM20H, Borgwaldt, Hamburg, Germany) as described in detail in ISO 4387:2019 [13] using 10 cigarettes per test in triplicate for each result. The ISO protocol used a 35 mL puff volume, 2 s puff duration and 60 s puff interval. Deviating from the ISO regime, the HCI smoking regime takes a 55 mL puff volume, 2 s puff duration and 30 s puff interval [15]. Ventilation holes in the cigarette tipping paper were fully blocked with the HCI holders (Borgwaldt, Hamburg, Germany) [15], but left unblocked for the ISO holders.

Total particulate matter was trapped on a glass fiber filter. The mass of the particulate matter was determined gravimetrically. Components collected on the glass fiber filter were extracted using isopropyl alcohol. After extraction, nicotine yields were determined by gas chromatography flame ionization detector (GC-FID) as described in ISO 10315:2021 [36]. CO was measured in the vapor phase of the cigarette smoke with a calibrated non-dispersive infrared (NDIR) gas analyzer (Borgwaldt, Hamburg, Germany) as described in ISO 8454:2007 [37].

### 2.6. Statistical Analyses

Analyses were conducted using Stata 16.1 (StataCorp, College Station, TX, USA) and graphed using Microsoft Excel for Microsoft 365 version 2405.

One-way ANOVAs followed by Bonferroni post-hoc tests were used to analyze physical cigarette measures and percent filter ventilation by color group. Bartlett’s test confirmed equal variances between groups. TNCO yield by color group and measurement protocol (ISO, HCI) was analyzed by 2-way ANOVAs. Levene’s test confirmed equal variances between groups. Univariate regression analyses were performed to examine each of four cigarette characteristics (filter ventilation, expanded tobacco, rod length and cigarette diameter) and TNCO yields using both ISO and HCI measurement protocols. Multivariate regression analyses were subsequently performed for each outcome using each measurement protocol to examine independent predictors including all cigarette characteristics in the model. As tobacco rod length and filter length were inversely proportional, only rod length was included as a proxy for tobacco volume when combined with cigarette diameter.

## 3. Results

### 3.1. Physical Properties of Cigarettes and Cigarette Tobacco

Gold cigarette variants had statistically significant shorter tobacco rods than blue and red cigarette variants (Figure 1a), (*F*(2,21) = 8.80, *p* = 0.0017). Gold cigarettes also had statistically significant longer filters (*F*(2,21) = 7.59, *p* = 0.0033) and tipping papers (*F*(2,21) = 8.80, *p* = 0.0017) than blue and red cigarettes. There was no statistically significant difference (*F*(2,21) = 1.85, *p* = 0.182) between the mean diameters of cigarettes in the gold (7.86 SD 0.076 mm), blue (7.79 mm SD 0.054 mm) and red groups (7.80 mm SD 0.084). Mean cigarette lengths appeared to be similar across color groups (gold 83.0 mm SD 0.341 mm, blue 83.2 mm SD 0.241 mm and red 83.2 mm SD 0.139 mm). These results suggest that blue and red cigarettes contain higher volumes of tobacco compared to gold cigarettes sold in Australia.

Displacement was used to indicate the presence and relative proportion of expanded tobacco compared to tobacco from the CM9 reference cigarette, which contains no expanded tobacco. As expected, tobacco from the 1RF6 reference cigarette (which contains some expanded tobacco), showed greater displacement (117%) relative to the CM9 reference. Tobacco from each of the 24 cigarette brands from the Australian market yielded greater than 117% relative to CM9, indicating that expanded tobacco was present in each brand (Figure 1b). Displacement in the samples ranged from 119% to 153%, with an overall mean of 133%. There were no statistically significant differences in mean displacement between color groups (*F*(2,21) = 0.04, *p* = 0.96, Figure 1b).

### 3.2. Filter Ventilation of Cigarettes

There was a statistically significant difference in the amount of filter ventilation by color variant (Figure 2; *F*(2,21) = 4.86, *p* = 0.018). A trend of reducing mean filter ventilation can be seen in Figure 2 where gold > blue > red. Bonferroni post-hoc tests indicated that the differences in filter ventilation between gold and blue (*p* = 1.000) and between blue and red variants (*p* = 0.145) were not statistically significant, but the difference between gold and red was statistically significant (*p* = 0.018). For five of the eight brands, the amount of filter ventilation of each color variant followed this pattern. For the other three brands, filter ventilation was similar across the three color variants.

Photographs of the tipping papers from deconstructed cigarettes in each brand revealed ventilation holes in all but one brand variant in the red group (Appendix A). This same red brand variant had 0% filter ventilation, as shown in Figure 2. Most tipping papers contained one row of holes, but the gold brand variant with the highest percent of filter ventilation had two rows of ventilation holes. Four brand variants contained an array of very small holes, with approximately >100 holes per filter. These numerous small holes did not lead to markedly higher or lower filter ventilation compared to other brand variants.

### 3.3. Tar, Nicotine and Carbon Monoxide Levels Measured by ISO and HCI Protocols

Two-way ANOVAs showed that overall, compared to the ISO protocol, the HCI protocol produced statistically significant higher measures for tar (*F*(1,21) = 1145.51, *p* < 0.0001), nicotine (*F*(2,21) = 739.79, *p* < 0.0001) and CO (*F*(2,21) = 1278.57, *p* < 0.0001) yields (Figure 3a–c). Overall HCI measurements were 268% higher than ISO measurements for tar, 245% higher than ISO measurements for nicotine, and 250% higher than ISO measurements for CO (Appendix A). There were statistically significant effects of color group on tar (*F*(2,21) = 10.54, *p* = 0.007), nicotine (*F*(2,21) = 4.71, *p* = 0.021) and CO (*F*(2,21) = 7.07, *p* = 0.005) yields (Figure 3a–c). While pairwise comparisons between each color group did not reach statistical significance, there was a trend for the mean tar and nicotine yields (Figure 3a,b) to increase from gold to blue to red brand variants when measured using both ISO and HCI protocols, and this was also apparent for CO when measured using the ISO but not the HCI protocol (Figure 3c).

### 3.4. Filter Ventilation and Tobacco Rod Length Are Associated with Tar, Nicotine and CO Yields

Filter ventilation was negatively associated with tar, nicotine and CO yields in univariate analyses when measured using the ISO but not the HCI protocol (Figure 4a–c).

Results of multiple regression analyses assessing whether cigarette characteristics (filter ventilation, expanded tobacco, tobacco rod length and cigarette diameter) predicted tar, nicotine or CO yields are reported in Table 1. When measured using the ISO protocol, filter ventilation was negatively associated with tar, nicotine and CO yields and tobacco rod length was positively associated with tar and nicotine yields. But when using the HCI protocol, tobacco rod length was the only characteristic showing a statistically significant positive association with tar, nicotine and CO yields.

## 4. Discussion

This study identified characteristics of popular cigarettes sold in Australia that are likely to underlie the misperception that some cigarettes have a reduced risk of harm. Filter ventilation, which confers an inherently misleading smoking experience, was universal among the cigarettes we tested, with the exception of one of the 24 products. Color names for brand variants were meaningful, to the extent that there was a gradient of filter ventilation, ranging from red cigarettes (least ventilation) to blue and then gold cigarettes (most ventilation). Filter ventilation was associated with smoking machine measured TNCO levels when the ISO protocol was used, but not when the HCI protocol was used. The presence of expanded tobacco was detected in all brand variants, but the relative amount was not associated with color variant or with TNCO measured by either ISO or HCI protocols. Gold cigarette variants also had slightly shorter tobacco rods, suggesting lower tobacco volume than blue or red variants, and tobacco rod length was positively associated with TNCO levels.

A pervasive myth among people who smoke is that cigarettes with smoother tasting smoke are less harmful and less addictive than those with harsher tasting smoke [2,4,5,18,19]. That filter ventilation makes smoke taste smoother without decreasing the risks of disease is well established [12,38,39]. Our results also show that filter ventilation likely remains a common feature of cigarettes sold in Australia and is associated with the color names used for brand variants, consistent with cigarettes sold in the United States [40,41].

In Australia and other countries, the restrictions on cigarette advertising, package labelling and plain packaging mean that the naming of cigarettes has been an important means of influencing user perceptions. Prior to Australia’s 2012 requirement for plain packaging, cigarette package colors were closely aligned to the ISO-measured TNCO yields from smoking machines, with increasing TNCO from white (most ventilated) to gold, blue, and then red (least ventilated) brand variants [42]. This implies that color names were an indication of the amount of filter ventilation used in cigarettes sold in Australia. While packs containing mentholated cigarettes were commonly colored green prior to 2012, the word “menthol” remains part of cigarette naming and “green” is rarely used in the names of contemporary cigarettes sold in Australia [26]. The current study shows that color variant names in Australia remain meaningful more than a decade after plain packaging was implemented, and convey misleading information. Variant names such as “gold” refer to cigarettes with relatively higher filter ventilation than “red” and this corresponds with the pattern of misperceptions of gold cigarettes as less harmful than the harshest cigarette variants such as reds [43,44]. Like filter ventilation [4], color variant names are therefore inherently misleading.

Australia has recently passed the Public Health (Tobacco and Other Products) Act 2023, the provisions of which will come into force over 2025 [45]. This legislation will ban brand variant color names, as well as variant names that highlight positive characteristics or refer to or imply no harm or reduced harm [46], so this may be an important step in reducing misperceptions. The legislation also standardizes filters in some important ways (e.g., no recessed filters or crushable flavored capsules) although it does not standardize or ban filter ventilation. Therefore, the compelling sensory experience of smoother smoke from more highly ventilated cigarettes will persist, irrespective of what these brand variants are called. It remains to be seen how tobacco companies will respond to the restrictions on the terms that can be used as part of brand variant names, for example, by potentially using alternative labeling practices not yet captured in legislation or by changing product additives or engineering, to maintain consumer satisfaction. It is worth noting that the new Australian legislation also bans most flavoring additives including menthol and its derivatives [46].

Interestingly, of the eight brands tested in this study, only five showed the trend of reducing percent filter ventilation from gold to red brand variants (Figure 2). These five brands consisted of all four of the brands manufactured by British American Tobacco and the single brand manufactured by Imperial. The two Philip Morris brands and one Richland Express brand did not follow this pattern; rather, the different color variants did not consistently differ in mean percent filter ventilation. One of these brands had an unusual form of filter ventilation in the tipping paper of all three color variants: an array of approximately 100 tiny holes (Appendix A). Overall, these findings are insufficient to generalize about filter ventilation across all brands sold in Australia. But the existence of some brands that do not meet the common pattern may signify a shift for some companies towards a medium level of filter ventilation regardless of color names. Tobacco companies may be anticipating regulation of filter ventilation, such as in Uruguay where single representation requirements limit the number of different ventilation levels within a brand family [47]. The industry may be experimenting with different types of ventilation or with adapting their customers to more similarity in strength in taste between color variants. The study of industry and consumer responses to policies that more comprehensively ban misleading variant names, such as Uruguay and Australia, or establish limits on emissions, provides important contextual information for nations that may ultimately seek to ban or standardize filter ventilation, or even ban filters altogether [48]. Researchers should monitor industry evolution of the relevant parameters of cigarette design to better understand how these influence emissions [16,49]. Research is encouraged to better understand the likely consumer response to a standard level of filter ventilation, or no ventilation at all [50,51].

Aside from differences in filter ventilation across different color variants, this study also showed that gold cigarettes had a shorter mean length of tobacco rod and longer filters and tipping papers than blue and red cigarettes. This result indicates that gold cigarettes are likely to have a lower volume of tobacco and higher volume of filter, as the cigarette length and diameter remained very similar across colors. The shorter length of tobacco rod is associated with lower TNCO yields, however it is not known whether this is due to less tobacco, the higher volume of filter, or to both characteristics. This finding of less tobacco volume in gold cigarettes appears to be previously unreported in the peer-reviewed literature. To further explore whether gold cigarettes sold in Australia had less tobacco, we referred to data from the most recent voluntary disclosures of ingredients from 2020, provided to the Australian Government by three tobacco companies [52,53,54]. The weight of tobacco as a proportion of the whole cigarette was calculated for all gold/yellow, blue and red cigarettes in these documents (excluding those with crushable filter capsules). For each of the three companies’ brands, the proportion of tobacco weight increased slightly from gold to blue to red (Appendix A). We conclude that it is common for gold cigarettes sold in Australia to have slightly less tobacco by weight than blue or red cigarettes.

These results show that overall, the HCI protocol yielded much higher levels of TNCO than the ISO protocol for this sample of cigarettes. This finding, combined with other studies that show that the ISO method underestimates human exposure, leads to the conclusion that the ISO protocol produces misleading results [20]. The higher yield of TNCO from the HCI protocol likely arises from differences in the puffing regime. For example, a higher frequency of puffs and blocking the ventilation holes in the HCI method mean an increased time in high-temperature combustion and less time in lower temperature smoldering between puffs, as well as increased airflow compared to the ISO regime [55]. Since the ISO measurement method does not account for compensatory smoking practices, the World Health Organization’s (WHO) Study Group on Tobacco Regulation (TobReg) has called for the HCI protocol to be the preferred machine-smoking method given that this more accurately emulates human behavior, such as by covering filter ventilation holes like a human does whilst smoking [14,56].

Irrespective of which smoking machine protocol is used, the measured TNCO yields are considered inaccurate indicators of the risk of disease caused by smoking, and therefore gold cigarettes should not be considered as less harmful on the basis of results from either protocol. To more accurately compare the toxicity of cigarette brand variants, TobReg recommends measuring a range of toxic chemicals in smoke as expressed per milligram of nicotine [55]. This approach moves towards a standardized comparison of different products, taking into account that people usually adjust their smoking behavior to gain a set amount of nicotine [55,57]. When the tar and carbon monoxide yields are expressed per milligram of nicotine per cigarette in the current data, there is very little difference between color variants and also little difference between results tested using ISO compared to HCI protocols (Appendix A). These data also support the notion that color variant names are misleading and that filter ventilation does not produce safer cigarettes.

Multivariate analyses showed that filter ventilation and tobacco rod length, but not cigarette diameter or expanded tobacco, were associated with TNCO levels in this sample when measured using the ISO protocol. But when the HCI protocol was used, only the tobacco rod length was associated with TNCO levels. These results indicate that the only reason that gold cigarettes yield slightly less TNCO once compensatory smoking is accounted for (i.e., using the HCI protocol), is that they contain slightly less tobacco. It should also be noted that these results measure TNCO per cigarette, and that there is evidence that some people who smoke gold cigarettes smoke more cigarettes per day. Since there was so little variation in the diameter of the cigarettes in this sample, any effect of diameter on TNCO would not be expected, so the influence of diameter on TNCO levels among the full range of cigarettes on the market cannot be ruled out.

Though it receives little attention compared to TNCO yields, the use of expanded tobacco has been touted by the tobacco industry and allies as reducing the harms of cigarettes [27]. The current results suggest that expanded tobacco remains a common feature of cigarettes sold in Australia. Unlike filter ventilation and tobacco rod length, the relative levels of expanded tobacco were not associated with color designations, nor with TNCO levels when measured by either smoking machine protocol. The current results are consistent with those from independent researchers showing no effect on TNCO levels [58] and also with results published by the tobacco industry [59], but at odds with claims of reduced harm by the tobacco industry [27]. The inclusion of expanded tobacco that made cigarettes weigh less was a likely consequence of taxation on the basis of weight, although weight-based taxation was discontinued in Australia in 1999 [29,60]. The benefits for the tobacco industry of maintaining expanded tobacco in contemporary cigarettes sold in Australia are unclear. However, we suspect that tobacco companies may have wanted to preserve the look and feel of tobacco to which consumers had become accustomed during decades of weight-based taxation. In addition, expanded tobacco might be expected to reduce industry production costs, since less unprocessed tobacco is needed per cigarette rod, assuming that any additional manufacturing steps are cost-efficient. Such a strategy would be especially advantageous to maintaining industry profits in nations such as Australia with relatively high tobacco taxation, where the final price includes a tax that is based on the cost of the product (goods and services or sales taxes).

This study has a number of limitations, particularly in that it is not a fully comprehensive analysis of the Australian cigarette market. The most commonly smoked color variants were chosen (gold, blue and red) but less popular brand variants that probably have very high levels of filter ventilation (e.g., white and silver) were not included. Other cigarette characteristics that may have made the smoke taste smoother, such as smoke pH and other additives were not measured as part of this study, and mentholated brands were excluded. The small number of brands tested (8) limited the power of the statistical analyses. A more comprehensive comparison of individual toxicants in color variants would be also more informative than measurements of tar, given that tar is a complex mixture of thousands of chemicals. As demonstrated previously, there may be an interaction between levels of toxicants, when some are increased while others are decreased in smoke from different cigarette brands [61]. A recommended approach is to compare a range of individual toxicants between samples [55].

## 5. Conclusions

Among the sample of Australian cigarette products tested, the extent of filter ventilation substantially affected TNCO yields using the ISO measurement protocol but not the HCI protocol that more closely mimics human smoking behavior. The length of the tobacco rod affected yields under both testing conditions. The level of expanded tobacco in the cigarette rod and the diameter of cigarettes did not influence TNCO emissions under either testing protocol for this sample. Using expanded tobacco may reduce industry costs, but more research and analysis is needed on any health implications of this apparently common practice. The results of this study confirm that the color variant names of cigarette brands, and the levels of filter ventilation they signify, are misleading, and likely continue to underpin misperceptions of reduced harm.

It will be important to monitor the extent to which Australia’s new tobacco control legislation, which bans variant names that use colors or other terms that imply reduced harms, may correct misperceptions. To optimize policy impact, we encourage the use of public communication campaigns and consumer education using misperception correction principles [62,63,64] to help people who smoke understand *why* smoother- or weaker-tasting cigarettes do not deliver lower emissions or reduce harm [65,66,67].

## Figures and Tables

**Figure 1 ijerph-22-00050-f001:**
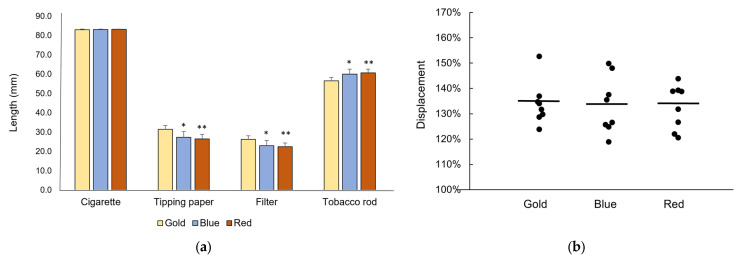
(**a**) Mean and standard deviation (*n* = 8 per group) of lengths of whole cigarettes, tipping paper, filters and tobacco rods by color group. Asterisks denote statistically significant difference to gold (* *p* < 0.05; ** *p* < 0.01); (**b**) Percent displacement relative to CM9 reference cigarette tobacco, as an indicator of the presence of expanded tobacco. Lines depict means for each group.

**Figure 2 ijerph-22-00050-f002:**
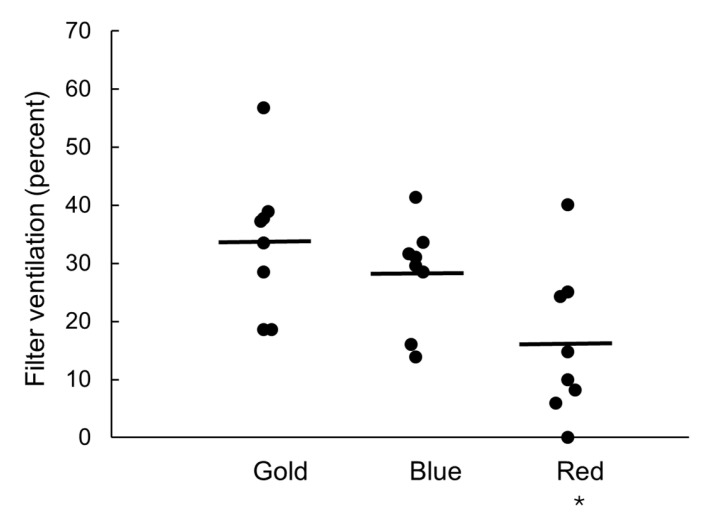
Filter ventilation of cigarette brand variants by color group (*n* = 8 brands per color group). Lines show the mean for each group. Asterisk denotes statistically significant difference to gold at *p* < 0.05.

**Figure 3 ijerph-22-00050-f003:**
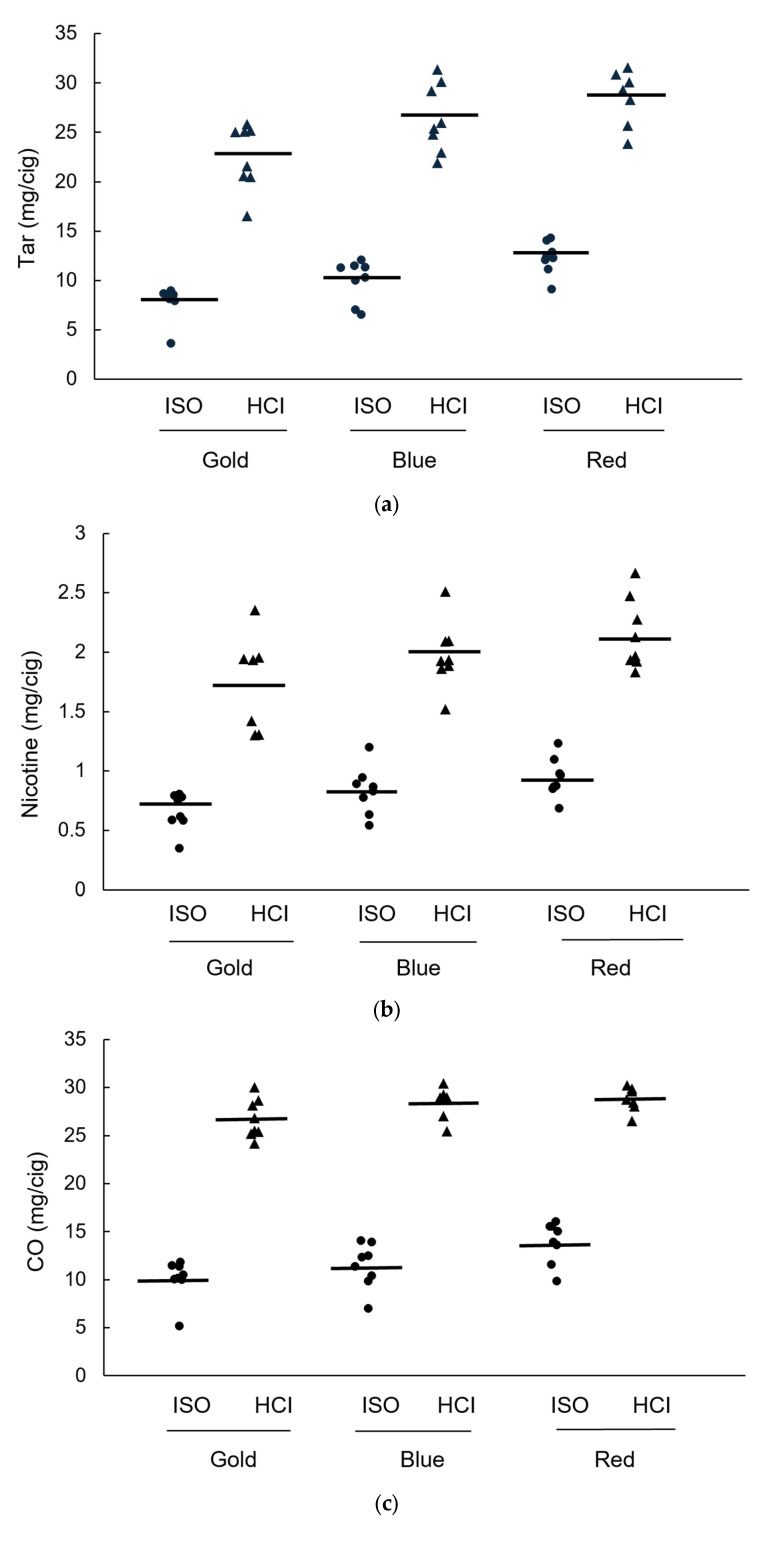
(**a**) Tar yield per brand; (**b**) Nicotine yield per brand; (**c**) CO yield per brand; by color variant group. Each group contains *n* = 8 brands, measured using the ISO (circles) or HCI protocols (triangles). Lines show the mean for each group, for each protocol.

**Figure 4 ijerph-22-00050-f004:**
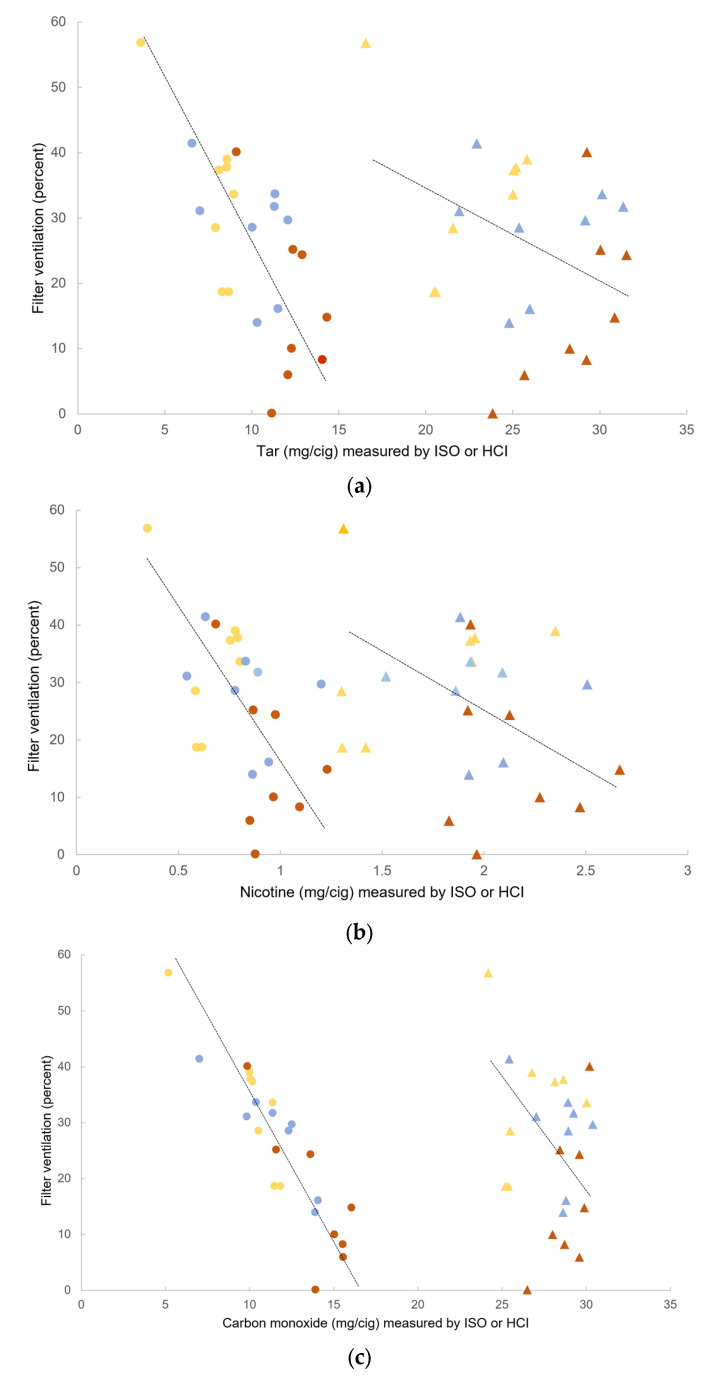
The relationship between filter ventilation and (**a**) tar, (**b**) nicotine and (**c**) carbon monoxide, as measured using the ISO protocol (circles) or the HCI protocol (triangles). Gold, blue and red colors refer to brand variant color. Lines are the regression line from the model, adjusted for other cigarette characteristics (see Table 1).

**Table 1 ijerph-22-00050-t001:** Results from multiple regression analyses with filter ventilation, expanded tobacco, tobacco rod length and cigarette diameter as predictors of tar, nicotine or CO yields using ISO or Health Canadian Intensive (HCI) protocol.

**Dependent Variables**	**Independent Variables**	**Coefficient (Standard Error)**	** *p* **
Tar (ISO protocol)	**Filter ventilation ***	**−0.111 (0.0262)**	**0.000**
	Expanded tobacco	2.50 (3.61)	0.496
	**Tobacco rod length**	**0.490 (0.126)**	**0.001**
	Diameter	4.66 (4.61)	0.325
Tar (HCI protocol)	Filter ventilation	−0.0323 (0.0502)	0.528
	Expanded tobacco	1.85 (6.91)	0.791
	**Tobacco rod length**	**1.12 (0.241)**	**0.000**
	Diameter	14.1 (8.82)	0.126
Nicotine (ISO protocol)	**Filter ventilation**	**−0.00515 (0.00240)**	**0.045**
	Expanded tobacco	0.646 (0.331)	0.066
	**Tobacco rod length**	**0.0475 (0.0115)**	**0.001**
	Diameter	0.378 (0.422)	0.382
Nicotine (HCI protocol)	Filter ventilation	−0.00155 (0.00528)	0.772
	Expanded tobacco	1.03 (0.727)	0.172
	**Tobacco rod length**	**0.0914 (0.0254)**	**0.002**
	Diameter	0.574 (0.929)	0.544
CO (ISO protocol)	**Filter ventilation**	**−0.166 (0.0204)**	**0.000**
	Expanded tobacco	3.76 (2.81)	0.196
	Tobacco rod length	0.168 (0.0979)	0.102
	Diameter	2.53 (3.59)	0.488
CO (HCI protocol)	Filter ventilation	−0.00363 (0.0289)	0.901
	Expanded tobacco	4.45 (3.98)	0.277
	**Tobacco rod length**	**0.405 (0.139)**	**0.009**
	Diameter	4.28 (5.08)	0.410

* Variables in bold are statistically significant at *p* < 0.05.

## Data Availability

The original data presented in this study are openly available in Zenodo at https://zenodo.org/records/14172220 (accessed on 16 November 2024).

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
