# Peer review of "The Effects of Filter Ventilation and Expanded Tobacco on the Tar, Nicotine and Carbon Monoxide Yields from Cigarettes Sold in Australia"

_ijerph, 2024, doi:10.3390/ijerph22010050_

Round 1
Reviewer 1 Report
Comments and Suggestions for Authors
TITLE The effects of filter ventilation and expanded tobacco on the tar, nicotine and carbon monoxide yields from cigarettes sold in Australia
AUTHORS Wendy R. Winnall et al
JOURNAL REF. IJERPH 3346730
The authors examine tar, nicotine and carbon monoxide (TNCO) yields in popular Australian cigarettes classified in three color variants "gold", "blue", "red". The main design characteristics of these variants (rod length, cigarette diameter, filter ventilation, "expanded" tobacco) are measured and described in full. The golden variant had on average shorter rods and greater filter ventilation, while ventilation was lower in the red variant. The authors compare yields obtained with the International Organization for Standardization (ISO) and Health Canada intense protocols. In the ISO protocol the greater ventilation of the "golden" variant produced lower TNCO yields. When tested with the Health Canada protocol yields increased significantly and minimized the effect of filter ventilation. The authors conclude that color variants associated with levels of filter ventilation are not associated with reduced TNCO yields, thus supporting incoming Australian regulation proposing to ban any association of harm reduction with color variants.
The article is clearly written and well explained. The authors provide (as far as I can tell) a good discussion on "expanded tobacco" and present newin formation on popular cigarette brands in Australia (for example the likely shorter rod length of the gold variant).
The authors provide a good account of various known issues in cigarette testing. For example, the inadequacy of machine testing cigarettes to represent exposure of human smokers. The broad consensus that the ISO protocol grossly underestimates human exposure, with the Health Canada protocol being a better proxy, though also far from ideal. The authors also stress the broad consensus that labels such as "low tar", "mild", etc, as well as various types of filtering do not represent a meaningful toxicity reduction, which in the end depends much more on smokers's puffing patterns, nicotine dependence and number of cigarettes per day, than on specific features of cigarette design.
However, the article needs to broaden the discussion of its limitations by addressing the following points:
1) The authors showed that TNCO yields of the three color variants significantly increase when passing from the ISO regime (35 mL puff volume) to the a more intense Health Canada (55 mL) protocol. More intense protocols are expected to produce higher yields per cigarette and less yield variation between brands, since they involve a large amount of smoke per cigarette. To avoid the dependence of yields on smoke volume per cigarette the World Health Organization Study Group on Tobacco Product Regulation (TobReg) proposal recommends for every puffing protocol and cigarette brand evaluating yields per mg of nicotine. Normalization to nicotine does not eliminate the difference of toxicant yields between puffing regimes, but allows for a more objective inter-brand standadised comparison between these two regimes. See figure 1 of
D M Burns et al, Mandated lowering of toxicants in cigarette smoke: a description of the World Health Organization TobReg proposal. Tobacco Control 2008;17:132–141. doi:10.1136/tc.2007.024158
As argued by Burns et al, the yield differences between different puffing regimes likely reflect different physical parameters that change from one regime to the other (combustion and smoldering temperature, airflow).
TobReg also highlights the limitations of assessing risks on the basis of TNCO yields, arguing that this restriction of toxicants may lead to misleading assessments, hence the suggestion of expanding the range of toxicants. TobReg also expresses caution on the limitations of looking at yields from individual toxicants, since the latter do interact. As shown in
Bill King, Ron Borland, Jeff Fowles. Mainstream smoke emissions of Australian and Canadian cigarettes. Nicotine & Tobacco Research Volume 9, Number 8 (August 2007) 835–844
The reduction of a given toxicant might be correlated with increase of another one ("risk swapping"), pointing out the case of TSNAs and polycyclic, aromatic hydrocarbons (PAHs).
RECOMMENDATION. The authors must provide a discussion of the limitations of their study in terms of the arguments expressed in the summary of TobReg in Burns et al and in King et al. I am not asking the authors to do any modification in their analysis and regulatory recommendations, but the discussion I am asking is important to provide an appropriate context to their study. Once the authors address these points the article can be published.
Author Response
Comment 1: The authors must provide a discussion of the limitations of their study in terms of the arguments expressed in the summary of TobReg in Burns et al and in King et al. I am not asking the authors to do any modification in their analysis and regulatory recommendations, but the discussion I am asking is important to provide an appropriate context to their study. Once the authors address these points the article can be published.
Response: We thank the reviewer for their insightful comments and recommendations that have improved our manuscript.
We agree with the reviewer that a discussion of these issues would strengthen our manuscript. We have added text and references to our manuscript regarding both issues at lines 426 to 446 and 484 to 490 (line numbers correspond to the simple markup view).
We have also added a figure to the supplementary information showing our tar and carbon monoxide yields expressed per milligram of nicotine (Figure S3).
Reviewer 2 Report
Comments and Suggestions for Authors
Winnall et al. investigated the effects of filter ventilation and expanded tobacco on the tar, nicotine and CO yields from cigarettes sold in Australia. While this is an interesting study, my major concern is that many references are rather old (published over 10 years ago). The Introduction reads more like a history book as a result. In the Discussion, it would be helpful to include comparisons with more recent studies. Some specific comments:
1. L167: In the numerator, add a space between “Filter” and “ventilation”.
2. Figure 1b: A box plot would be more appropriate. I have the same suggestion for Figures 2 and 3. Also, “% displacement” is informal. Please remove '%' from the y-axis label. The same suggestion applies to other Figures/Tables in the manuscript/supplement where applicable.
3. Figure 2: Add “statistically” before “significant difference”. This should be applied elsewhere in the manuscript as well.
4. L371-372: Reference is missing.
5. L460: The word “surmise” sounds informal. Please rephrase.
Author Response
Comment 1: Winnall et al. investigated the effects of filter ventilation and expanded tobacco on the tar, nicotine and CO yields from cigarettes sold in Australia. While this is an interesting study, my major concern is that many references are rather old (published over 10 years ago). The Introduction reads more like a history book as a result.
Response 1: We thank the reviewer for their insightful comments and suggestions that have improved our manuscript.
We have made numerous changes to the introduction, including:
- removed some of the dates and older information in order to state only the necessary background to understand why filter ventilation and expanded tobacco are in use today,
-replaced many references with more recent research.
These changes can be seen on lines 44, 47 to 48, 67 to 68, 79 and 89 to 90 of the new manuscript (all line numbers are correct for the simple markup view).
Comment 2: In the Discussion, it would be helpful to include comparisons with more recent studies.
Response 2: Our discussion does refer to some older studies. However, in the parts of the discussion where we have made comparisons of our results to those of others, these studies are all quite recent.
We note that comparisons between our results and others are made on:
- lines 352 to studies from 2014 and 2018,
- line 367 to studies from 2022 and 2014,
- line 401 to a study from 2018,
- lines 464 to 467 to studies from 2023, 2003 and 2007.
We believe that we have made comparisons to the most appropriate studies and cannot improve the manuscript by citing any that are more recent.
Comment 3: L167: In the numerator, add a space between “Filter” and “ventilation”.
Response 3: completed, see line 163 in the new version of the manuscript.
Comment 4: Figure 1b: A box plot would be more appropriate. I have the same suggestion for Figures 2 and 3.
Response 4: We respectfully disagree that a boxplot is appropriate for the data in figures 1b, 2 and 3. Since the data were analysed using parametric tests (one-way and two-way ANOVAs) then a figure that shows the mean is the most appropriate for displaying the data. Boxplots show the median and interquartile ranges, which are appropriate for data analysed using non-parametric tests. We have chosen to show each datapoint in each group to give the reader a complete view of the dataset, as well as a line for the mean, which is appropriate for data analysed using parametric tests. Please note that for figure 1a, the data were so tightly clustered that such an approach was not useful here, so we have used a bar graph shown the mean and standard deviation instead.
Comment 5: Also, “% displacement” is informal. Please remove '%' from the y-axis label. The same suggestion applies to other Figures/Tables in the manuscript/supplement where applicable.
Response 5: We have replaced the % symbols in figures 1b, 2, 4a to 4c, S2 and Table S2 in the new manuscript and new supplementary information.
Comment 6: Figure 2: Add “statistically” before “significant difference”. This should be applied elsewhere in the manuscript as well.
Response 6: We have made these changes as suggested at numerous places in the manuscript. See the highlighted words on lines: 197, 199, 200, 214, 219, 223, 228, 234, 246, 250 and 324 of the new manuscript.
Comment 7: L371-372: Reference is missing.
Response 7: A reference has been added, please see line 370.
Comment 8: L460: The word “surmise” sounds informal. Please rephrase.
Response 8: we have replaced “can surmise” with “suspect”, please see line 471 of the new manuscript.
Round 2
Reviewer 2 Report
Comments and Suggestions for Authors
The revised manuscript has improved significantly. I have no more comments.